# Upconversion Nanoparticles Intercalated in Large Polymer Micelles for Tumor Imaging and Chemo/Photothermal Therapy

**DOI:** 10.3390/ijms241310574

**Published:** 2023-06-24

**Authors:** Polina A. Demina, Kirill V. Khaydukov, Gulalek Babayeva, Pavel O. Varaksa, Alexandra V. Atanova, Maxim E. Stepanov, Maria E. Nikolaeva, Ivan V. Krylov, Irina I. Evstratova, Vadim S. Pokrovsky, Vyacheslav S. Zhigarkov, Roman A. Akasov, Tatiana V. Egorova, Evgeny V. Khaydukov, Alla N. Generalova

**Affiliations:** 1Federal Scientific Research Center «Crystallography and Photonics» of the Russian Academy of Sciences, 119333 Moscow, Russia; polidemina1207@yandex.ru (P.A.D.); haidukov_11@mail.ru (K.V.K.); alexandraii@mail.ru (A.V.A.); ivan_krylov@bk.ru (I.V.K.); vzhigarkov@gmail.com (V.S.Z.); roman.akasov@gmail.com (R.A.A.); 2Shemyakin–Ovchinnikov Institute of Bioorganic Chemistry of the Russian Academy of Sciences, 117997 Moscow, Russia; 3Institute of Physics, Technology, and Informational Systems, Moscow State Pedagogical University, 119435 Moscow, Russia; stepanov_me@mail.ru (M.E.S.); mesarycheva@gmail.com (M.E.N.); irina.evs02@gmail.com (I.I.E.); tatvladegorova@gmail.com (T.V.E.); 4N.N. Blokhin National Medical Research Center of Oncology, Ministry of Health of Russia, 115478 Moscow, Russia; babaevagulyalek@gmail.com (G.B.); varaksa.pavel@yandex.ru (P.O.V.); vadimpokrovsky@gmail.com (V.S.P.); 5Research Institute of Molecular and Cellular Medicine, Peoples’ Friendship University of Russia (RUDN University), 117198 Moscow, Russia; 6Scientific Center for Translation Medicine, Sirius University of Science and Technology, 354340 Sochi, Russia; 7Institute of Molecular Theranostics, Sechenov University, 119991 Moscow, Russia

**Keywords:** upconversion nanoparticles, polymer micelles, bioimaging, photothermal therapy, combined therapy, thermosensitive polymer, theranostic agent

## Abstract

Frontiers in theranostics are driving the demand for multifunctional nanoagents. Upconversion nanoparticle (UCNP)-based systems activated by near-infrared (NIR) light deeply penetrating biotissue are a powerful tool for the simultaneous diagnosis and therapy of cancer. The intercalation into large polymer micelles of poly(maleic anhydride-alt-1-octadecene) provided the creation of biocompatible UCNPs. The intrinsic properties of UCNPs (core@shell structure NaYF_4_:Yb^3+^/Tm^3+^@NaYF_4_) embedded in micelles delivered NIR-to-NIR visualization, photothermal therapy, and high drug capacity. Further surface modification of micelles with a thermosensitive polymer (poly-N-vinylcaprolactam) exhibiting a conformation transition provided gradual drug (doxorubicin) release. In addition, the decoration of UCNP micelles with Ag nanoparticles (Ag NPs) synthesized in situ by silver ion reduction enhanced the cytotoxicity of micelles at cell growth temperature. Cell viability assessment on Sk-Br-3, MDA-MB-231, and WI-26 cell lines confirmed this effect. The efficiency of the prepared UCNP complex was evaluated in vivo by Sk-Br-3 xenograft regression in mice for 25 days after peritumoral injection and photoactivation of the lesions with NIR light. The designed polymer micelles hold promise as a photoactivated theranostic agent with quattro-functionalities (NIR absorption, photothermal effect, Ag NP cytotoxicity, and Dox loading) that provides imaging along with chemo- and photothermal therapy enhanced with Ag NPs.

## 1. Introduction

Recent efforts in nanotheranostics have focused on the development of personalized approaches to cancer diagnosis and therapy, where nanoparticles play a prominent role, as they can be tailored for required biochemical properties [1,2]. A significant advantage of nanoparticles (NPs) is associated with the ability to create multifunctional particulate agents that provide comprehensive anticancer treatment [3]. In particular, NPs can act as drug carriers in combination with visualization functionalities and stimuli-responsive therapy, monitored drug localization, and controlled on-site treatment, thereby minimizing side effects [4,5]. The critical point concerns the integration of imaging and therapy, which can both improve and enhance their effects [6].

Upconversion nanoparticles (UCNPs) have attractive optical features and are considered promising candidates for theranostics [7]. UCNPs are composed of rare earth inorganic nanomaterials and exhibit anti-Stokes luminescence under NIR-light excitation, which underlies new instrumental techniques for minimally invasive in vivo and in vitro imaging in the biological transparency window without autofluorescence [8]. UCNPs of various sizes, shapes, and phases [9] have prominent visualization advantages, including narrow emission peaks, large anti-Stokes shifts, photostability, and long lifetimes [10]. Moreover, UCNPs provide high spatial resolution, which depends on the nature of the host matrix, the type and concentration of dopants (ions of rare earth elements), additional ions (iron, silver, gold, etc.), crystalline phase, structure (core, core/shell, core/active shell/shell), size, shape, and surface modification [8,11]. In addition, UCNPs can be employed in magnetic resonance, X-ray, photoacoustic imaging, computed tomography, positron emission tomography, etc. [12]. These imaging functionalities aim to predict and monitor therapeutic outcomes in real time.

UCNPs can exhibit photothermal functionality owing to the dopant, which is a rare earth ion (Yb^3+^) with a large light absorption cross-section [13]. Nanoparticles accumulated in the tumor tissue can provide local overheating under laser irradiation at 970–980 nm. This phenomenon is crucial for photothermal therapy (PTT) when tissue hyperthermia (above 42 °C) leads to tumor regression and subsequent cell death with minimal side effects on normal cells [14]. Additionally, the photothermal properties of UCNPs can be enhanced by complex formation with NPs possessing a larger absorption cross-section (gold, silver, carbon, etc.) [15,16]. Silver nanoparticles (Ag NPs) relate to this NP group due to the strong surface plasmon resonance absorption, which is beneficial for PTT. The combination of Ag NPs with UCNPs can strongly enhance the complex absorbance at low irradiation intensity. UCNP assembly with plasmonic NPs determines the special functionality of UCNPs.

The large surface area of nanoparticles provides other unique functionalities for theranostics since numerous approaches to surface modification have been developed. These approaches address the formation of UCNP coating with various functionalities, including reactive groups, the hydrophilic/lipophilic balance, adsorption properties, diverse thicknesses, densities, environmental resistance, and colloidal stability [17,18,19].

In particular, the surface coating can offer stimuli-responsive features endowed by the polymers used to modify the UCNP surface [20,21]. For example, stimuli-responsive polymers can change the solubility accompanied by conformational transformations in response to environmental conditions. The deposition of such polymers on the nanoparticle surface can alter UCNP optical and targeting properties arising from the conformational change of the polymers in a controlled manner when exposed to external stimuli. The effects of thermal stimuli attract much attention [22,23]. A prime example of a thermoresponsive polymer is poly-N-vinylcaprolactam (PVCL) [24,25], suffering the reversible conformation transition from a coil to a globule at 32–34 °C, which is called the low critical solution temperature (LCST). This peculiar PVCL behavior at LCST underlies the controlled drug release. UCNPs are beneficial for modification with thermosensitive polymers as this allows NIR light-guided, dosed, and localized drug release. The coating can provide a high drug loading for enhanced therapeutic effects on site and protects fragile drugs against hydrolytic degradation and fluorescence quenching. At the same time, drugs can affect the polymer shells, dictating the need for precise control of drug loading [26,27,28]. The incorporation of anticancer drugs (e.g., doxorubicin, Dox) into the UCNP polymer coating reduces systematic toxicity, drug resistance, and severe side effects for the patient, such as cardiotoxicity; it also enhances the local drug therapeutic dose in comparison with the free chemotherapeutic agent [29]. Moreover, in this configuration of the UCNP complex, additional functionality is expected due to the cytotoxic properties of Ag NPs and their simultaneous action with Dox, which shows a higher antitumor activity than the individual components [30].

A combination of the aforementioned functionalities in a single platform inspires us to design a construction responsible for image-guided therapy under external stimuli. Herein, we demonstrate a multifunctional theranostic complex activated by NIR light, exhibiting imaging and therapeutic effects simultaneously. The complex is based on UCNPs intercalated into large micelles of poly(maleic anhydride-alt-1-octadecene), modified with PVCL, and loaded with Dox for NIR-to-NIR visualization and chemo/photothermal therapy enhanced by Ag NPs.

## 2. Results and Discussion

### 2.1. UCNP-Based Multifunctional Complex Formation

The UCNPs with core/shell structure NaYF_4_:Yb^3+^/Tm^3+^@NaYF_4_ were synthesized by a modified solvo-thermal method, as previously described in [31]. TEM images and selected area electron diffraction patterns of as-synthesized UCNPs confirmed the mean diameter to be 30 nm and the presence of a hexagonal β-phase, which is the most favorable for the upconversion process (Figure 1a). The photoluminescence spectrum is presented in Appendix A. For detailed information about the UCNP photoluminescence mechanism, we recommend the reviews [32,33,34].

The developed approach to multifunctional UCNP complex design included four steps: 1—hydrophilization of hydrophobic UCNPs with amphiphilic poly(maleic anhydride-alt-1-octadecene) (PMAO); 2—thermosensitive polymer shell formation; 3—embedding the therapeutic agents; 4—decoration with Ag NPs. The general schematic illustration of rational design is presented in Figure 1.

The UCNPs were synthesized in organic media and stabilized with hydrophobic oleic acid (OA), demonstrating a lack of biocompatibility that requires modification with polymers for biomedical applications. The first step is the hydrophilization of the initially hydrophobic UCNPs in order to impart colloidal stability to them in an aqueous medium. Furthermore, the modification should ensure the deposition of stimuli-sensitive polymers on the surface of the complex. UCNPs were modified with amphiphilic polymer PMAO. Nanoparticles were mixed with PMAO in an organic solvent providing the PMAO hydrocarbon moiety interaction with the oleate ligand on the surface of UCNPs due to the action of hydrophobic forces. After transfer to water, PMAO can self-assemble into so-called large spherical micelles [35] with several UCNPs intercalated in the micelle core. The core is formed by hydrophobic moieties, while hydrophilic moieties are exposed to the aqueous phase [36,37]. Micelles were crosslinked with diamine to form stable polymer particles (see Figure 1b). Spontaneous hydrolysis of PMAO anhydride groups leads to carboxyl group formation on the surface, ensuring further adsorption of the stimuli-sensitive polymer. The success of UCNP–PMAO modification was confirmed by the negative zeta-potential −33 mV and Fourier-transform infrared (FTIR) spectroscopy (Appendix A). This modification approach ensures colloidal stability of NPs, reduces toxicity, prevents polymer removal from the surface, creates a template for multifunctionalization and targeting, and maintains high fluorescence intensity by protecting UCNPs from water.

The dosed and localized drug release can occur in the case of stimuli-responsive properties of UCNP complexes, for example, properties that change with temperature. In the second step (Figure 1c), we formed a thermosensitive polymer shell from poly-N-vinylcaprolactam (PVCL), structural formula see in Appendix A, capable of a reversible change in conformation from coil to globule at LCST in the range of 32–34 °C. The peculiar PVCL behavior at LCST can tune the release in such a manner: by loading drugs into the PVCL coil conformation and releasing the encapsulated drugs from the shrunk globule when the temperature rises above the LCST. This polymer is an excellent candidate due to its LCST and biocompatibility, which has not been thoroughly demonstrated for abundantly-used thermosensitive poly(N-isopropyl acrylamide) (PNIPAM) [38]. At the temperature above LCST, PVCL changes its conformation, which is related to the destruction of bonds with the associated water molecules and transforms into the globule conformation. On the particle surface, this polymer behavior manifests itself as the coating shrinkage. This PVCL property can tune the local drug release to avoid side effects. The complex of PVCL with UCNP–PMAO formed as a result of hydrogen bond formation between the C=O of PVCL amide groups and carboxyl groups of PMAO [39]. In order to confirm the PVCL coating formation, we analyzed FTIR spectra of the prepared UCNP–PMAO–PVCL complexes (see S2) and demonstrated the change in zeta-potential (−6 mV). The obtained complexes remained stable for a long time and were unaffected by electrolyte (PBS, pH 7.2).

The third step included the drug embedding as the chemotherapeutic functionality of the complex. We employed widely used Dox, which is an anticancer drug model with cytostatic effects. Owing to the positive charge and hydrophilic properties, Dox can easily penetrate the PVCL coating and hydrophilic area of UCNP–PMAO micelles, interacting electrostatically with the carboxyl groups. This ensures the effective Dox-loading of the complex. The concentration of Dox loaded into the UCNP–PMAO complex, determined spectrophotometrically, was about 0.32 mg Dox in relation to 1 mg UCNPs, while the concentration of Dox encapsulated into UCNP–PMAO–PVCL was lower (0.21 mg in relation to 1 mg). The loading decrease in UCNP–PMAO–PVCL appears to be caused by screening the charged carboxyl groups of PMAO by PVCL coating, where Dox is entrapped due to non-polar interactions.

Finally, UCNP–PMAO–PVCL complexes were decorated with Ag NPs, which enhanced the photothermal properties of UCNPs and exhibited cytotoxicity toward cancer cells. Ag NPs were formed on the complex surface in situ by coordinating silver ions with atoms having a lone electron pair (O, N) followed by ion reduction. Briefly, silver nitrate salt as a precursor of Ag NPs was added to the complexes, which were purified after incubation from unabsorbed Ag^+^. Then Ag^+^ ions were transferred to Ag^0^ using a “fast” reducing agent—sodium borohydride. In order to maintain the balance between the rate of nucleation and growth of Ag NPs, various concentrations of silver nitrate salt and reducing agent were tested. The decoration of the complex surface with Ag NPs was confirmed by EDX elemental mapping (Figure 1f), as well as by the increase in the colloidal stability. This effect refers to the NP property to increase the colloidal stability of dispersions, as in the case of Pickering emulsions.

### 2.2. Complex Characterization

One of the complex functionalities is related to temperature-driven drug release. The thermosensitive properties of UCNP–PMAO–PVCL were studied by hydrodynamic size measurement using dynamic light scattering analysis (DLS) in the temperature range from 28 to 42 °C (Figure 2a). Heating up to 32 °C led to a gradual decrease in the complex diameter, then dramatically decreased in the range of 32–37 °C. The fastest size change occurred in the range of 32–34 °C, which is associated with the aforementioned hydration-dehydration of PVCL. This dependence of the diameter on the temperature indicates the thermosensitive features of the designed multifunctional complex.

The study of drug release from Dox-loaded UCNP–PMAO–PVCL complexes was carried out in acidic (pH 5.3) and neutral (pH 7.2) media at 25 °C and 37 °C (Figure 2b) by acquiring the Dox absorption in the supernatants after centrifugation. The free Dox absorbance spectrum is shown in Appendix A. The PVCL coating of UCNP–PMAO micelles at rising temperatures gave the increase of Dox concentration in the dispersion medium, which prevailed in the case of UCNP–PMAO samples. The Dox release at pH 5.3 and 37 °C was about twice as high as at pH 7.2 at the same temperature. This is important because the pH is acidic in the tumor and neutral in normal tissue [40]. The pK_b_ of Dox is 8.2 [41], which indicates higher amino group protonation in Dox at low pH, providing a positive charge [42], while the carboxyl groups of PMAO become partially uncharged. This underlies the raised drug release for UCNP–PMAO at acidic pH. The increased Dox positive charge makes the non-polar interactions with PVCL weaker, providing higher drug release, given the contribution of Dox released from PMAO. These results reveal the pH-dependent character of drug release from multifunctional UCNP complexes, which can be used to fine tune the chemotherapy on site with minimal side effects.

The photothermal functionality of the designed complex is associated with resonant NIR light absorption in UCNPs. The absorption cross-section of Yb^3+^ ions is sufficient to cause local overheating of a surrounding media with cytotoxic effects on cells and tissues [13]. Photothermal functionality can be enhanced through drug encapsulation into the complex. The 980 nm laser light irradiation of the Dox-loaded UCNP–PMAO–PVCL complex induced UCNP heating, shrinkage of PVCL, and drug release (Figure 2c). The UCNP–PMAO–PVCL complex was decorated with Ag NPs to increase the heating rate. The heating rate difference between the complex decorated with Ag NPs and the Ag-free complex became evident in the first 20 s of NIR light exposure (Figure 2d). Additionally, we acquired changes in the absorption spectrum of Dox after Ag NP addition, manifested by an increase and a blue shift in the Dox absorption peak at 490 nm, indicating a change in the chemical environment that can be addressed to Dox–Ag NP assembly (see S4).

### 2.3. Cytotoxic Activity

The viability of cells in the presence of the designed complexes was evaluated using MTT assay after 24 and 72 h incubations (Figure 3a–c). It was found that UCNP–PMAO–PVCL complexes (without Dox, Ag NPs) were of low cytotoxicity with IC_50_ values even higher than 0.2 mg/mL in the case of human immortalized WI-26 fibroblasts, which confirmed the biocompatibility of the PMAO–PVCL coating. The low cytotoxicity of free UCNPs is correlated with the literature data, and this is usually discussed as one of the key advantages of UCNP-based systems [12]. However, the cytotoxicity of UCNPs mainly depends on the coating material [43,44], and the development of biologically inert coating makes it possible to increase safety both in vitro and in vivo [19,45]. The toxicity of the proposed UCNP-based complexes could be discussed as comparatively low since, in general, the cytotoxicity of UCNPs appears at concentrations 50–200 µg/mL [43,44,46]. Complexes containing Ag NPs and Dox/Ag NPs were toxic to cells after the 24 h incubation (IC_50_ 3.0–4.5 µg/mL), while the Dox-loaded complex demonstrated lower toxicity (IC_50_ was 6.8 µg/mL for the most sensitive Sk-Br-3 cells). This could be explained by the cytotoxic effect of Ag NPs [47,48]. However, after the 72 h incubation, the toxicity of Dox-loaded UCNP–PMAO–PVCL increased sharply, especially for WI-26 and MDA-MB-231 cell lines, which could be explained by the release of Dox from the complex. It should be noted that the release of Dox was sustained while Ag NPs released from the Dox-free complex probably did not occur due to the formation of a coordination bond with the complex coating. It is also necessary to note that after 24 h incubation, the toxicity of complex containing both Ag NPs and Dox was higher than the toxicity of complexes loaded with only Ag NPs- or only Dox-loaded complexes for all cell lines. This allows one to discuss the additive or even synergistic cytotoxic effect of their combination. Note that the same effects for Dox and Ag NPs are reported in the literature [30,49].

The designed complex can be visualized in two photoluminescent channels: anti-Stokes luminescence centered at 800 nm under 980 nm laser excitation (see S1) and Stokes luminescence centered at 590 nm under blue light excitation. The release of Dox from the nanocomplex was confirmed by confocal microscopy (Figure 3d–f). After 24 h incubation, the Dox fluorescence was mainly in the bright dots within the cytoplasm, which could be lysosomes with entrapped complexes. However, after 72 h incubation, the dot-like fluorescence decreased, which is explained by the pH-dependent character of the drug release from the complex under an acidic (pH 4.5–5.0) lysosomal media, followed by the accumulation of Dox in the cell nuclei.

Additionally, Dox toxicity is often discussed as delayed because of DNA intercalation resulting in cytostatic action [50]. Thus, in our control experiment, the IC_50_ for free Dox changed from 8.04 µM, 10.32 µM, and 1.14 µM (24 h) to 0.56 µM, 0.83 µM, and 0.47 µM (72 h) for MDA-MB-231, Sk-Br-3, and WI-26, respectively (Appendix A).

### 2.4. Antitumor Effect on Xenografts

The evaluation of the antitumor effect of the complex was accompanied by the whole-animal visualization (Figure 4a) using a custom-developed epi-luminescence imaging system capable of detecting 800 nm emission from UCNPs under 980 nm laser excitation [51]. Sk-Br-3 cells 5 × 10^6^ were administrated into the flanks of Balb/c nude mice to induce tumor formation (for details, see M&M section). The experiment was started on the 10^th^ day after inoculation when the average volume of tumors reached 70 ± 10 mm^3^. The complexes were administrated peritumorally (50 µL suspension, 0.8 mg/mL, single dose). Local overheating was employed after 24 h of the complex administration. NIR laser irradiation without complex injection was used as a negative control. The actual temperature of mice was monitored by a Xenics Gobi-384-GigE-7098 camera in real time. Tumors were treated with 980 nm pulsed laser light for an average of 3 min until the temperature difference between the tumor and untreated tissue reached 4 °C (Figure 4b). The rate of tissue heating depended on the injected sample. In the case of UCNP–PMAO–PVCL, the temperature was the same as in control (without UCNP) in the first few seconds but started to increase by an average of 1 °C after 1 min from the start of NIR laser treatment. As expected, the maximum heating rate was detected for the complex decorated with Ag NPs.

After 24 h of the NIR laser treatment, three mice were sacrificed for histological analysis. In the histological preparation, the tumor node was represented by atypical epithelial cells of various shapes and sizes with a light cytoplasm and a dark nucleus along the periphery for the control group. Pathological forms of mitosis were observed; mitotic activity was high (3–4 cells per screen). Along the periphery of the tumor, nested structures were formed in the connective tissue component. The connective tissue component was moderately developed and was represented by collagen and elastin fibers with fibroblasts. The vascular bed was poorly developed. The histological picture corresponded to a growing tumor (Appendix A). A similar histological picture was demonstrated for the control group treated with a 980 nm laser (Figure 4c).

The tumors treated with UCNP–PMAO–PVCL–Dox–Ag NP complex without 980 nm laser irradiation did not demonstrate any histological differences from the control groups (Figure 4e). At the same time, both NIR-irradiated groups were characterized by alterations in the histological picture. The tumors treated with the UCNP–PMAO–PVCL–Dox complex and NIR light showed areas of necrosis and hemorrhage, a moderate macrophage reaction, and a decrease in mitotic activity (0–1 cells per screen, necrotic cells 1–2 per screen) (Figure 4d). Treatment with the UCNP–PMAO–PVCL–Dox–Ag NP complex and 980 nm laser irradiation revealed necrosis areas and a moderate macrophage reaction, as well as a mild sclerotic process. Mitotic activity was 1–2 cells per screen (Figure 4f).

Tumor growth was evaluated throughout the observation period. The in vivo NIR-induced treatment efficacy was assessed by measuring the tumor volumes over a period of 25 days. The results are presented in Figure 5a,b. The additional control of tumor growth after the free Dox injection is shown in Appendix A. Most importantly, the 980 nm laser irradiation can further enhance the efficacy of therapy using UCNP–PMAO–PVCL–Dox and UCNP–PMAO–PVCL–Dox–Ag NP complexes. By the end of the experiment, subcutaneous xenograft volumes and the tumor growth indices in the group receiving UCNP–PMAO–PVCL–Dox and UCNP–PMAO–PVCL–Dox–Ag were significantly lower than the corresponding values in the control groups. Specifically, as shown in Figure 5c, the tumor growth inhibition (TGI) value on the 25th day of observation for the UCNP–PMAO–PVCL–Dox complex was 59% (*p* = 0.01), while for the UCNP–PMAO–PVCL–Dox–Ag NP complex it was 67% (*p* = 0.02). Neither the UCNP–PMAO–PVCL–Dox–Ag NP complex without 980 nm laser irradiation nor the control group treated with 980 nm laser showed statistically significant TGI against control. The single dose injection of the complex without/with 980 nm laser irradiation was well-tolerated with a loss of body weight of no more than 8% (no statistical difference between control and treatment groups). The most effective therapy with high inhibition of tumor growth was achieved after 980 nm laser irradiation in the presence of a multifunctional complex containing either Dox or Dox–Ag NPs. Thus, the combined chemo- and photothermal therapy using a UCNP-based platform reduced the Sk-Br-3 tumor growth rate by ~5 times compared to the untreated control.

## 3. Materials and Methods

### 3.1. Materials

Following materials were purchased from Sigma-Aldrich (St. Louis, MO, USA) and used without further purification: sodium chloride, phosphate buffered saline pH 7.2 (PBS), poly(maleic anhydridealt-1-octadecene, PMAO (Mn 30,000), hexamethylenediamine, poly(N-vinylcaprolactam) (Mn 30,000), tannin, silver nitrate salt, ammonia, sodium borohydride, doxorubicin. Chloroform was of analytical grade and purchased from Sigma-Aldrich.

Optical data were evaluated using Evolution 201 spectrophotometer (Thermo Scientific, Waltham, MA, USA). Photoluminescence spectra were acquired by spectrofluorometer Fluorolog 3 (Horiba Jobin Yvon, Longjumeau, France). The particle size and zeta-potential of UCNP samples in PBS buffer (pH 7.2) were detected by Zetasizer Nano ZS (Malvern Panalytical, Malvern, UK) at 25 °C. The actual temperature of samples and mice was measured by Xenics Gobi-384-GigE-7098 camera in real time. UCNP complexes were studied by transmission electron microscopy (TEM) in bright-field mode, high-resolution TEM mode, high-angle annular-DF–scanning transmission electron microscopy (HAADF–STEM), and energy-dispersive X-ray analysis using an electron microscope (Tecnai Osiris, Thermo Fisher Scientific) operated at an accelerating voltage of 200 kV.

### 3.2. Methods

#### 3.2.1. UCNP Intercalation in PMAO Micelles

In order to prepare hydrophilic UCNP nanocomplexes, UCNP dispersion (40 µL, 20 mg/mL) in chloroform was added to amphiphilic PMAO (130 µL, 8 mg/mL) in chloroform. The mixture was stirred, sonicated for 5 min, and incubated for 2 h under stirring at room temperature. After 1 h incubation, 3 µL hexamethylenediamine as a crosslinking agent was added. Then, the mixture was added dropwise under vigorous stirring and sonication to 1 mL H_2_O. Solvent was evaporated under heating for 40 min, and the mixture was centrifuged at 20.8 kg for 10 min with buffer addition (this procedure was repeated three times to remove free PMAO), and then was dispersed in 1 mL PBS buffer, pH 7.2.

#### 3.2.2. UCNP–PMAO–PVCL Preparation

Next, 1 mL UCNP–PMAO (0.8 mg/mL) was centrifuged at 20.8 kg for 10 min; supernatant was replaced with 1 mL of poly(N-vinylcaprolactam) (PVCL, 4 mg/mL) aqueous solution. The mixture was incubated with stirring at room temperature for 1 h, then 0.002 mg of tannin was added as a crosslinking agent and stirred for another 10 min. The mixture was centrifuged for 10 min at 20.8 kg; the supernatant was replaced with 1 mL of PBS.

#### 3.2.3. Doxorubicin Loading into UCNP–PMAO–PVCL

Doxorubicin (0.2 mL, 2 mg/mL) was added to 1 mL (0.8 mg/mL) of UCNP–PMAO–PVCL in PBS and stirred for 30 min. Then, to remove an excess of Dox, centrifugation was carried out for 10 min at 20.8 kg, then the supernatant was replaced by 1 mL of PBS.

#### 3.2.4. UCNP–PVCL Decoration with Ag NPs

A precursor of Ag NPs in the form of silver nitrate salt 0.16 mL (2.5 mg/mL) was added to 1 mL of UCNP–PMAO–PVCL in water in the presence of ammonia (5 μL). After incubation at room temperature for 1 h, the excess of non-absorbed silver ions was removed by centrifugation for 10 min at 20.8 kg. Finally, reducing agent sodium borohydride (20 μL, 1 mg/mL) was added. After 24 h incubation, the probe was centrifuged for 10 min at 20.8 kg; the supernatant was replaced with 1 mL of PBS.

#### 3.2.5. Dynamic Light Scattering Measurement

Hydrodynamic sizes of the UCNP–PMAO–PVLC complex were detected by Zetasizer Nano ZS (Malvern Panalytical, Malvern, UK) at 28–42 °C temperature range with 2 °C step. The dispersion was diluted with water to obtain the concentration required for the light scattering experiments according to the manual and then poured into a cuvette. The measurement was carried out 15 min after each temperature point.

#### 3.2.6. Dox Release Study

The degree of Dox release from UCNP–PMAO–PVLC complex was evaluated spectrophotometrically by measuring the optical absorption of free Dox in the supernatant. The samples were centrifuged at 20.8 kg for 10 min, and the supernatant was collected and measured at the range of wavelength of 300–700 nm. The concentration of released Dox was calculated from the calibration graph, and the concentration of Dox added at the stage of complex preparation.

#### 3.2.7. Cytotoxicity Studies

Human breast adenocarcinoma cells (MDA-MB-231 or Sk-Br-3) or immortalized human fibroblasts WI-26 were seeded onto a 96-well plate (10^4^ cells per well) and incubated overnight. Then, UCNPs suspensions were sterilized under UV (30 min) and added to the cells, and plates were incubated at 37 °C for 24 h or 72 h. Cell viability was evaluated by MTT assay at 24 h after UCNPs adding. The viability of control (intact) cells was taken as 100%.

#### 3.2.8. Confocal Microscopy

Human breast adenocarcinoma Sk-Br-3 cells were seeded onto 8-well glass slides (5 × 10^4^ cells per well) and incubated overnight. Then, 5 µg/mL of UCNP–PMAO–PVCL–Dox nanocomplex was added to the cells for 24 h and 72 h. After the incubation, the cells were washed three times with PBS (pH 7.2), fixed in 4% PFA, and additionally stained with Hoechst 33342 (1 µg/mL, 15 min) to visualize cell nuclei. The fluorescence was evaluated using Leica confocal TCS SPE system using 405 nm and 543 nm lasers.

#### 3.2.9. Animals, Visualization, and Evaluation of Antitumor Activity

Female Balb/c nude mice for antitumor activity studies (6–8 weeks old, average 20–22 g body weight) were obtained from the N.N. Blokhin National Medical Research Center of Oncology (Moscow, Russia).

In vivo optical imaging of tumor with assistance of UCNPs was demonstrated in a subcutaneous xenograft model using the human breast cancer cell line Sk-Br-3; 3.5 × 10^6^ cells were suspended in 0.2 mL DMEM (Gibco) and transplanted subcutaneously into the right flank of mice. Once the mean tumor volume reached ∼70 mm^3^, animals were randomly assigned to 5 groups, 5 mice per group. In therapeutic groups, mice were treated once with peritumoral injection 50 µL of UCNPs (0.8 mg/mL) in PBS (pH 7.2) modified with PMAO–PVCL, PMAO–PVCL with Dox, PMAO–PVCL with Ag NPs and PMAO–PVCL with both Dox and Ag NPs and 50 µL of free Dox (2 µg/mL). Mice were housed in a pathogen-free environment under controlled conditions of light and humidity and received food and water ad libitum.

Local overheating was demonstrated 24 h after injection. NIR laser irradiation without UCNP nanocomplex injection was used as a control. The actual temperature of mice was measured by Xenics Gobi-384-GigE-7098 camera in real time. Mice were treated with the pulsed NIR laser (power 225 mW, intensity 0.45 W cm^−2^) for an average of 3 min until the temperature difference between the untreated area and treated tumor reached 4 °C.

The distribution of PL signals was observed in vivo using home-built visualization system [43]. The epi-luminescent images were obtained at 60 min after injection of the NPs.

Tumor diameters were measured twice per week, and the tumor volume was estimated as (length × width × height × π)/6. The relative tumor volume (RTV) was calculated using the following formula: RTV = (tumor volume on measured day)/(tumor volume on day 0). On day 29, the tumor growth inhibition ratio (TGI, %) was calculated using the following formula: TGI (%) = [1 − (RTV of the treated group)/(RTV of the control group)] × 100 (%).

The tolerability of the therapy was determined by measuring body weight, monitoring appearance daily, and autopsy findings.

#### 3.2.10. Histological Assays

After animal euthanasia by CO_2_ inhalation 48 h after injection, tumor biopsies were fixed with 10% formalin for 72 h, embedded in paraffin blocks, and histological sections 4 µm thick were made using microtome (thermo) according to the standard procedure. The sections were then stained with hematoxylin and eosin according to the standard protocol. Tumor response was assessed in several fields of view.

#### 3.2.11. Statistical Analysis

Statistical analysis of tumor growth was performed using the one-tailed Student’s *t*-test in Statistica 10.0 software. The mean values and standard deviation (SD) were represented on the graphs as error bars. A statistical value of *p* was determined between the control and each treatment group. *p* < 0.05 was regarded as statistically significant.

### 3.3. Key Insights

To date, cancer is one of the most high-risk and fatal diseases in the world. Despite significant progress in cancer therapy, it remains a complex medical problem, especially cancers of the breast, liver, prostate, pancreas, and brain. Conventional cancer treatments, such as chemotherapy, surgical excision, and radiotherapy, have drawbacks, including toxic and severe side effects, drug resistance, low specific targeting ability, and limited efficacy. Recent research efforts have focused on developing personalized approaches to cancer diagnosis and therapy, where nanoparticles play a prominent role due to their diverse, easily adjustable surface and bulk properties. A significant advantage of nanoparticles is associated with the ability to create multifunctional particulate nanoscale agents that provide a complex effect.

There are many reports in the literature about a multifunctional “theranostic” UCNP nanoplatform, including different functionalities in a single hybrid system to provide—along with optical visualization—magnetic resonance, X-ray, photoacoustic imaging, computed tomography, etc. [12]. However, the improved efficacy of cancer treatment was shown to be achieved by integration with chemotherapy or photodynamic or photothermal therapy, exhibiting the disease treatment effect. To date, the critical point concerns the roles of imaging and therapy, which are often performed separately from each other. Revealing therapeutic effects through imaging, and implementing the therapeutic effects to enhance imaging, is challenging [6].

Most publications that reported on the multifunctional complex of UCNPs with Dox as a model drug indicate the weak development of this field. However, UCNPs have great potential as drug delivery carriers since the complex appeared to be more effective in treating cancer in vivo than the free drug [52]. It is worth noting that the polymer shell has not yet been optimized for high drug loadings, though reports of porous silica coating were very promising. Drugs can interfere with the properties of polymer shells; therefore, a fine balance is required between drug-loading capacity and changes in the polymer shell [27,28].

Herein, we demonstrate a theranostic agent based on UCNPs, exploiting all four aforementioned functionalities: NIR absorption, photothermal effect, complex formation with Ag NPs, surface modification providing a thermosensitive polymer coating, and drug encapsulation. Effective loading of Dox as a model drug was realized due to UCNP modification by intercalation into large micelles of poly(maleic anhydride-alt-1-octadecene) without affecting the polymer properties. The UCNP complex with Dox possessing thermosensitive properties inherited from the polymer and Ag NPs, synthesized in situ by silver ion reduction, showed a cytotoxic effect at the temperature of cell growth (37 °C). In addition, cell viability significantly decreased after prolonged incubation (72 h) with the UCNP complex containing Ag NPs compared to the complex without Ag NPs, probably due to the synergetic action of Dox and Ag NPs, described in the literature. Moreover, Dox exhibits fluorescence detectable by conventional devices (not required irradiation at 980 nm), such as a confocal microscope, facilitating the assessment of a multifunctional UCNP effect on cell death. In other words, therapy effects associated with the destruction of cancer cells were evaluated through imaging. We demonstrated the efficiency of the UCNP complexes for PTT in vivo after exposure to NIR light therapy under imaging guidance. Peritumoral injection of the UCNP complex and maintaining the temperature in the site under pulse NIR irradiation with visual control ensured a therapeutic effect, manifested in significant tumor degradation. Thus, multifunctional UCNP complexes containing Dox with or without Ag NPs gained the most effective therapy with significant tumor growth inhibition after NIR irradiation. These results, indicating a superior effect of the simultaneous exploitation of Dox and Ag NPs, are in good agreement with the data of cell experiments. The developed multifunctional UCNP complex holds promise as a photoactivated theranostic agent delivering imaging along with chemotherapy and photothermal therapy.

## 4. Conclusions

We demonstrated the approach to fabricate a multifunctional thermosensitive complex based on UCNPs for simultaneous imaging and chemo- and photothermal therapy under NIR light irradiation. Effective loading of Dox as a model drug was realized due to UCNP modification using intercalation in poly(maleic anhydride-alt-1-octadecene) large micelles without affecting the polymer properties. The thermosensitive properties of the UCNP complex, inherited from the thermosensitive polymer and Ag NPs, regulate the therapeutic effect of Dox, enhancing the cytotoxic effect at the cell growth temperature (37 °C). In addition, cell viability significantly decreased after prolonged incubation (72 h) with multifunctional UCNPs as compared to the Ag-free complex, probably, due to the synergetic action of Dox and Ag NPs. We demonstrated in vivo, in a mice xenograft model, the efficiency of prepared UCNP complexes for Sk-Br-3 tumor regression. Peritumoral injection of the UCNP complex and maintaining the temperature on site under 980 nm laser pulse irradiation with visual control ensured a five-fold decrease in the tumor growth rate compared to the untreated control. This scenario only occurred for the Dox-loaded complexes without and with Ag NPs.

## Figures and Tables

**Figure 1 ijms-24-10574-f001:**
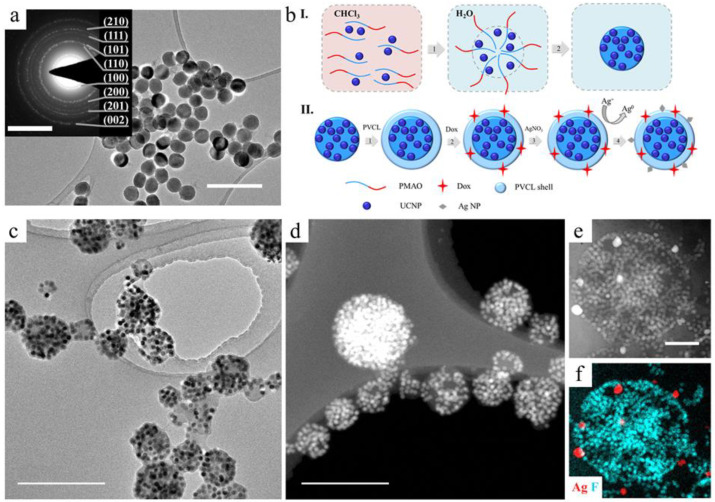
(**a**) TEM images of as-synthesized core/shell NaYF_4_:Yb^3+^/Tm^3+^@NaYF_4_ UCNPs (scale bar - 100 nm) and selected area electron diffraction pattern of UCNPs on insert (scale bar - 5 nm^−1^). (**b**) Schematic illustration of the rational design of multifunctional UCNP complexes: I.—step 1: PMAO hydrocarbon moiety interaction with oleate ligand on the surface of UCNP in organic solvent; PMAO self-assembly into large spherical micelles with several UCNPs intercalated in the micelle core after transfer into water; crosslinking of large spherical micelle with diamine to form stable polymer particles (UCNP–PMAO); II.—step 2: UCNP–PMAO modification with PVCL; step 3: embedding the therapeutic agents; step 4: Ag^+^ adsorption on the UCNP–PMAO–PVCL surface, followed by reduction; (**c**) TEM and (**d**) HAADF–STEM image of UCNP–PMAO–PVCL complexes. Scale bar is 500 nm; (**e**) HAADF–STEM image and (**f**) EDX elemental mapping of UCNP–PMAO–PVCL containing in situ-formed Ag NPs. Scale bar is 200 nm.

**Figure 2 ijms-24-10574-f002:**
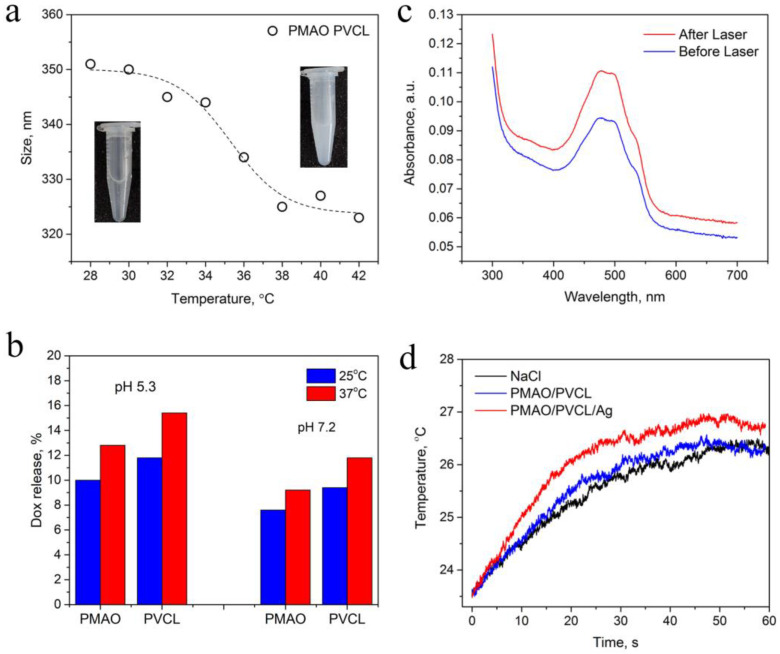
(**a**) Size decrease of UCNP–PMAO–PVLC complex under heating acquired by DLS. (**b**) Release of Dox from UCNP–PMAO and UCNP–PMAO–PVCL complexes in response to temperature (25 and 37 °C) at different pH (5.3 and 7.2). (**c**) Release of Dox from complex in response to 980 nm pulsed laser treatment. (**d**) Heating rate of NaCl solution containing UCNP–PMAO–PVCL in comparison with UCNP–PMAO–PVCL–Ag NPs under 980 nm laser irradiation (intensity 0.5 W/cm^2^, complex concentration 0.4 mg/mL).

**Figure 3 ijms-24-10574-f003:**
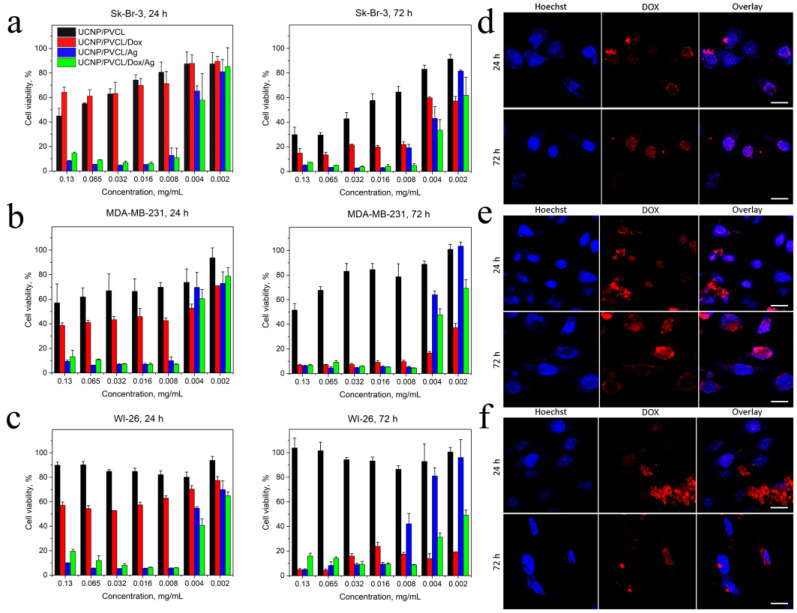
Cell viability of (**a**) human breast adenocarcinoma Sk-Br-3 cells. (**b**) Human breast adenocarcinoma MDA-MB-231 cells. (**c**) Human WI-26 fibroblasts under NP treatment, 24 h and 72 h incubation, MTT assay. The data are presented as mean ± SD. Confocal microscopy of (**d**) Sk-Br-3 cells, (**e**) MDA-MB-231 cells, (**f**) WI-26 fibroblasts incubated with PMAO–PVCL Dox nanocomplexes within 24 h and 72 h. Blue is for Hoechst 33342 (cell nuclei), red is for Dox. Scale bar is 20 µm.

**Figure 4 ijms-24-10574-f004:**
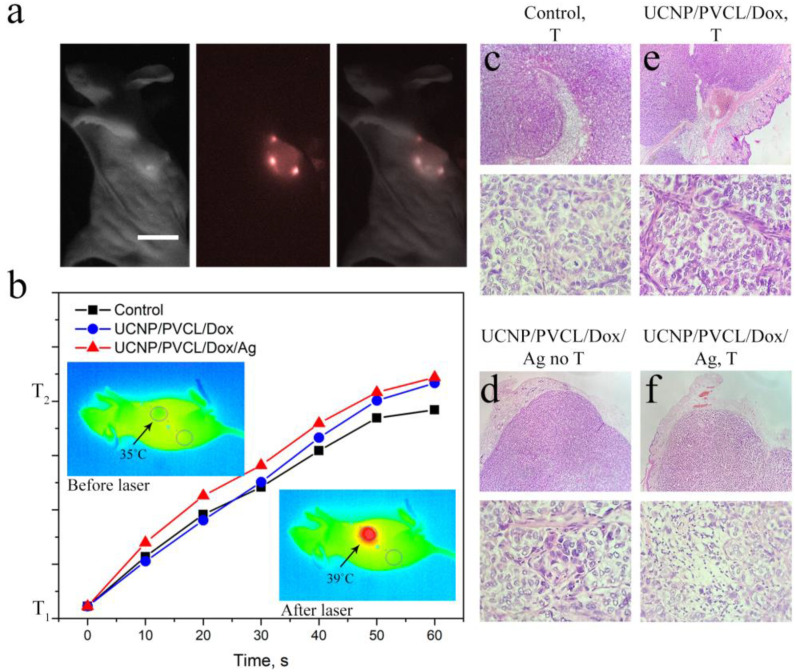
(**a**) In vivo images taken after 1 h UCNP–PMAO–PVCL complex injection: bright-field (left), epi-luminescent (center), and overlay (right) images. Scale bar 1 cm. (**b**) The rate of tumor heating depending on the injected complex; inserts—temperature images of mice before and after 980 nm laser treatment demonstrate tumor overheating. (**c**) Histological images of the tumor tissue sections stained with hematoxylin and eosin after the 980 nm laser treatment, negative control group; (**d**) after UCNP–PMAO–PVCL–Dox complex injection with 980 nm laser treatment; (**e**) after UCNP–PMAO–PVCL–Dox–Ag NP complex without 980 nm laser treatment; and (**f**) UCNP–PMAO–PVCL–Dox–Ag NPs with 980 nm laser treatment. Top images—magnification 40×; bottom images—magnification 400×.

**Figure 5 ijms-24-10574-f005:**
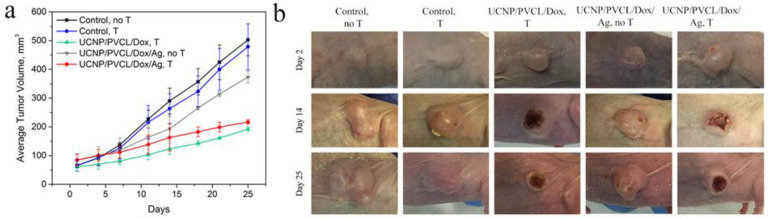
(**a**) Tumor volume regression in the Balb/c nude mice with Sk-Br-3 human breast cancer xenografts post treatment. Tumor volume at each time point is shown in line graphs. *n* = 5 mice/group. *p* < 0.05. Mean ± SD. (**b**) A time-lapse series of the bright-field photographs of the Sk-Br-3 tumor area of control and treated groups.

## Data Availability

Available upon request.

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
