# Peer review of "Upconversion Nanoparticles Intercalated in Large Polymer Micelles for Tumor Imaging and Chemo/Photothermal Therapy"

_ijms, 2023, doi:10.3390/ijms241310574_

Round 1

Reviewer 1 Report

In their paper "Upconversion nanoparticles intercalated in large polymer micelles for tumor imaging and combined chemo/photothermal therapy" Demina et al investigate the preparation and use of novel Up-conversion nanoparticles (UCNPs) as efficient approach to fabricate multifunctional thermos-sensitive complex based on UCNPs for simultaneous imaging, chemo- and photothermal therapy under NIR-light irradiation. The paper is well written, deals with an important bio-application and presents an elegant approach. The experiments are well designed and executed. Overall, I consider that the content of this paper is proper for the IJMS:

Comments:

1.       The UCNPs were coated with PMAO and then modified with PVCL for good stability of the NPs in aqueous dispersion. Authors claims the particles stability however no data were presented to support their claim. DLS or z-potential in H2O, PBS, etc versus time is recommended.

2.       Figure 1.: The average particles (UCNP) are shown with about ~250 nm in SEM images based on the scale bar presented in Figure 1a-d, please include scale bar in Figure 1-e & f as well.

3.       To correlate and prove no changes in the particle size, please add a scale bar to Figure 4a and correlate to Figure 1 (i.e., the average particle size).

4.       Please include a paragraph and comment on the “Advantages of the UCNP”- presented here,   section, and compare your particles system against other measurement systems.

5.       Figure 2d: the y-axis is missing, please correct.

6.       References need to be checked and corrected:

Some titles are all caped, and some are not. e.g.,:

15. Generalova, A.N.; Oleinikov, V.A.; Khaydukov, E. V One-dimensional necklace-like assemblies of inorganic nanoparticles: Recent advances in design, preparation and applications. Adv Colloid Interface Sci 2021, 297, 102543.

17. Chung, Y.-C.; Yang, C.-H.; Lee, R.-H.; Wang, T.-L. Dual Stimuli-Responsive Block Copolymers for Controlled Release Triggered by Upconversion Luminescence or Temperature Variation. ACS omega 2019, 4, 3322–3328.

Please correct:

Ref.4: Fang, C.; Zhang, M. Nanoparticle-based theragnostics: Integrating diagnostic and therapeutic potentials in nanomedicine. J Control release Off J Control Release Soc 2010, 146, 2–5.

Author Response

Dear Reviewer,

Thank you for your consideration of our manuscript. The authors’ team is grateful to your comments, which helped to improve the content of the manuscript. We have made the corrections in the manuscript.

Please, find our reply in the file attached. The comments are in Italic, followed by our response in Regular front.

Sincerely yours,

Evgeny V. Khaydukov

Reviewer 2 Report

The manuscript reports on upconversion nanoparticles with silver decorations for chemo- photothermal and cytotoxic treatment of cancerous cells. It is well written and logically organized into sections. Please note many figures in the supplementary cannot be clearly seen. The results are interesting and the nanoparticles are promising for multiple functions in theranostics. There are however several results that are lacking the required characterization evidence in order to draw conclusions and require a more concise introduction in the manuscript. The quality of the images should also be improved. Detailed comments are attached in a pdf document.

Minor editing of English language required

Author Response

(The authors gave the same response as above.)

Round 2

Reviewer 2 Report

The authors applied in-depth revisions according to the concerns of the review that improved the presentation of the manuscript and can be published.